



**Atmospheric Chemistry and Physics Discussions**

**Unexpected enhancement of ozone exposure and health risks during National Day in China**
Peng Wang[1, 2], Juanyong Shen[3], Men Xia[1], Shida Sun[4], Yanli Zhang[2], Hongliang Zhang[5,6], Xinming Wang[2]
[1]Department of Civil and Environmental Engineering, The Hong Kong Polytechnic University, Hong Kong,
China
[2]State Key Laboratory of Organic Geochemistry and Guangdong Key Laboratory of Environmental
Protection and Resources Utilization, Guangzhou Institute of Geochemistry, Chinese Academy of Sciences,
Guangzhou, China
[3]School of Environmental Science and Engineering, Shanghai Jiao Tong University, Shanghai, China
[4]Tianjin Key Laboratory of Urban Transport Emission Research, College of Environmental Science and
Engineering, Nankai University, Tianjin, China
[5]Department of Environmental Science and Engineering, Fudan University, Shanghai, China
[6]Institute of Eco-Chongming (IEC), Shanghai, China
*Correspondence to:* Yanli Zhang (zhang_yl86@gig.ac.cn)
**Abstract**
China is confronting increasing ozone ($O_3$) pollution that worsens air quality and public health. Extremely
$O_3$ pollution occurs more frequently under special events and unfavorable meteorological conditions. Here
we observed significantly elevated maximum daily 8-h average (MDA8) $O_3$ (up to 98 ppb) during the
Chinese National Day Holidays (CNDH) in 2018 throughout China, with a prominent rise by up to 120%
compared to the previous week. Air quality model shows that increased precursor emissions and regional
transport are major contributors to the elevation. In the Pearl River Delta region, the regional transport
contributed up to 30 ppb $O_3$ during the CNDH. Simultaneously, aggravated health risk occurs due to high
$O_3$, inducing 33% additional deaths throughout China. Moreover, in tourist cities such as Sanya, daily
mortality even increases by up to 303%. This is the first comprehensive study to investigate $O_3$ pollution
during CNDH at national level, aiming to arouse more focuses of the $O_3$ holiday impact from the public.




**Graphical abstract**

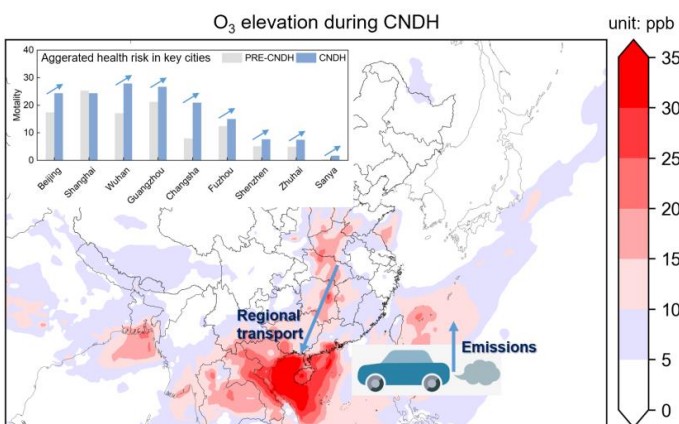


**1. Introduction**
Tropospheric ozone ($O_3$) has become a major air pollutant in China especially in urban areas such as
the North China Plain (NCP), Yangtze River Delta (YRD) and Pearl River Delta (PRD) in recent years,
with continuously increasing maximum daily 8-h average (MDA8) $O_3$ levels (Fang et al., 2019;Li et al.,
2019;Lu et al., 2018;Liu et al., 2018a). Exacerbated $O_3$ pollution aggravates health risks from a series of
illnesses such as cardiovascular diseases (CVD), respiratory diseases (RD), hypertension, stroke and
chronic obstructive pulmonary disease (COPD) (Liu et al., 2018a;Li et al., 2015;Brauer et al.,
2016;Lelieveld et al., 2013;Wang et al., 2020b). In China, the annual COPD mortality due to $O_3$ reaches up
to $8.03 \times 10^4$ in 2015 (Liu et al., 2018a).
$O_3$ is generated by non-linear photochemical reactions of its precursors involving volatile organic
compounds (VOCs) and nitrogen oxides ($NO_x$) (Sillman, 1995;Wang et al., 2017). The VOCs/$NO_x$ ratio
determines $O_3$ sensitivity that is classified as VOC-limited, transition and $NO_x$-limited, which controls $O_3$
formation (Sillman, 1995;Sillman and He, 2002;Cohan et al., 2005). Also, regional transport was reported
as an important source of high $O_3$ in China (Gao et al., 2016;Wang et al., 2020a;Li et al., 2012a). For
instance, Li et al. (2012b) showed that over 50% of surface $O_3$ was contributed from regional transport in
the PRD during high $O_3$ episodes.
$O_3$ concentration shows different patterns between holidays and workdays (Pudasainee et al., 2010;Xu
et al., 2017). Elevated $O_3$ has been observed during holidays in different regions resulted from changes in
precursor emissions related to intensive anthropogenic activities (Tan et al., 2009;Chen et al., 2019;Tan et
al., 2013;Levy, 2013). In China, most studies focused on the Chinese New Year (CNY) to investigate long-
term holiday effect on $O_3$ in southern areas (Chen et al., 2019). However, the Chinese National Day





Holidays (CNDH), a nationwide 7-day festival, is less concerned. Xu et al. (2017) reported that the $O_3$
production was influenced by enhanced VOCs during CNDH in the YRD based on in-situ observations.
Previous studies mainly paid attention to developed regions/cities without nationwide consideration. In
addition, the national $O_3$-attributable health impact during CNDH is also unclear. Consequently, a
comprehensive study on $O_3$ during the CNDH is urgently needed in China.
In this study, we used observation data and a source-oriented version of the Community Multiscale
Air Quality (CMAQ) model (Wang et al., 2019a) to investigate $O_3$ characteristics during the CNDH in 2018
in China. Daily premature death mortality was evaluated to determine health impacts attributed to $O_3$ as
well. We find a rapid increase by up to 120% of the observational MDA8 $O_3$ from previous periods to
CNDH throughout China, which is attributed to increased precursors and regional transport. This study
provides in-depth investigation of elevated $O_3$ and its adverse health impacts during CNDH, which has
important implications for developing effective control policies in China.
**2. Methods**
**2.1 The CMAQ model setup and validation**
The CMAQ model with three-regime (3R) that attributed $O_3$ to $NO_x$ and VOCs based on the $NO_x$-
VOC-$O_3$ sensitivity regime was applied to study the $O_3$ during CNDH in China in 2018. The regime
indicator R was calculated using Eq. (1):

$$R = \frac{P_{H_2O_2} + P_{ROOH}}{P_{HNO_3}} \tag{1}$$

where $P_{H_2O_2}$ is the formation rate of hydrogen peroxide ($H_2O_2$); $P_{ROOH}$ is the formation rate of organic
peroxide (ROOH), and $P_{HNO_3}$ is the formation rate of nitric acid ($HNO_3$) in each chemistry time step. The
threshold values for the transition regime are 0.047 ($R_{ts}$, change from VOC-limited to transition regime)
and 5.142 ($R_{te}$, change from transition regime to $NO_x$-limited regime) in this study (Wang et al., 2019b).
The formed $O_3$ is entirely attributed to $NO_x$ or VOC sources, when R values are located in $NO_x$-limited
($R > R_{te}$) or VOC-limited ($R < R_{ts}$) regime. In contrast, when R values are in the transition regime
($R_{ts} \leq R \leq R_{te}$), the formed $O_3$ is attributed to both $NO_x$ and VOC sources. Two non-reactive $O_3$ species:
$O_3\_NO_x$ and $O_3\_VOC$ are added in the CMAQ model to quantify the $O_3$ attributable to $NO_x$ and VOCs,
respectively. The details of the 3R scheme are described in Wang et al. (2019b).
A domain with a horizontal resolution of $36 \times 36$ km$^2$ was applied in this study, covering China and its
surrounding areas (Fig. S1). Weather Research and Forecasting (WRF) model version 3.9.1 was used to
generate the meteorological inputs, and the initial and boundary conditions were based on the FNL
reanalysis data from the National Centers for Environmental Prediction (NCEP). The anthropogenic





emissions in China are from the Multiresolution Emission Inventory for China (MEIC,
http://www.meicmodel.org/) version 1.3 that lumped to 5 sectors: agriculture, industries, residential,
power plants, and transportation. The monthly profile of the anthropogenic emissions was based on
Zhang et al. (2007) and Streets et al. (2003) as shown in Table S1 to represent the emissions changes
between September and October. Emissions from other countries were from MIX Asian emission
inventory (Li et al., 2017). Open burning emissions were from the Fire INventory from NCAR (FINN)
(Wiedinmyer et al., 2011), and biogenic emissions are generated using the Model of Emissions of Gases
and Aerosols from Nature version 2.1 (MEGAN2.1) (Guenther et al., 2012). The Integrated Process Rate
(IPR) in the Process Analysis (PA) tool in the CMAQ model was applied to quantify the contributions of
atmospheric processes to $O_3$ (Gipson, 1999) (details see Table S2).

The simulation period was from 24 September to 31 October in 2018 and divided into three intervals:

PRE-CNDH (24-30, September), CNDH (1-7, October) and AFT-CNDH (8-31, October). In this study, a
total of 43 cities that includes both megacities (such as Beijing and Shanghai) and popular tourist cities
(such as Sanya) were selected to investigate the $O_3$ issue during CNDH in 2018 in China (Table S3).
Locations of these cities cover developed (such as the YRD region) and also suburban/rural regions (such
as Urumqi and Lhasa in western China), which provides comprehensive perspectives for this study (Fig.
S1).

All the statistics results of the WRF model are satisfied with the benchmarks except for the GE of

temperature (T2) and wind speed (WD) went beyond the benchmark by 25% and 46%, respectively (Table
S4). The WRF model performance is similar to previous studies (Zhang et al., 2012;Hu et al., 2016) that
could provide robust meteorological inputs to the CMAQ model. The observation data of key pollutants
obtained from the national air quality monitoring network (https://quotsoft.net/air/, more than 1500 sites)
were used to validate the CMAQ model performance. The model performance of $O_3$ was within the criteria
with a slight underestimation compared to observations, demonstrating our simulation is capable of the $O_3$
study in China (Table S5).

**2.2 Health impact estimation**

The daily premature mortalities due to $O_3$ from all non-accidental causes, CVD, RD, hypertension,

stroke and COPD are estimated in this study. The $O_3$-related daily mortality is calculated based on
Anenberg et al. (2010) and Cohen et al. (2004). In this study, the population data are from all age groups,
which may induce higher daily mortality than expected (Liu et al., 2018a). In this study, the daily
premature mortality due to $O_3$ is calculated from the following Eq. (2)  (Anenberg et al., 2010;Cohen et
al., 2004) :

$\Delta M = y_0[1 - exp(-\beta \Delta X)]Pop$                    (2)


where $\Delta M$ is the daily premature mortality due to $O_3$; $y_0$ is the daily baseline mortality rate, collected from
the China Health Statistical Yearbook 2018 (National, 2018); $\beta$ is the concentration-response function
(CRF), which represents the increase in daily mortality with each 10 μg m$^{-3}$ increase of MDA8 $O_3$
concentration, cited from Yin et al. (2017); $\Delta X$ is the incremental concentration of $O_3$ based on the threshold
concentration (35.1ppb) (Lim et al., 2012;Liu et al., 2018a); $Pop$ is the population exposure to $O_3$, obtained
from China's Sixth Census data (Fig. S2) (National Bureau of Statistics of China, 2010). The daily $y_0$ and
$\beta$ values for all non-accidental causes, CVD, RD, hypertension, stroke and COPD are summarized in Table
S6.

**3. Results and Discussions**
**3.1 Observational $O_3$ in China during CNDH**

MDA8 $O_3$ levels have noticeably risen during the 2018 CNDH based on observations, from 43 ppb

(PRE-CNDH) to 55 ppb (CNDH) among selected cities (Figure 1a and Table S3). The largest increase of
MDA8 $O_3$ (up to 56%) is observed in South China (Fig. 1b). The PRD region has recorded 49 % of MDA8
$O_3$ increase and in most PRD cities (such as Shenzhen and Guangzhou), number of exceeding days is as
high as 5~7 days during the 7-day CNDH, which contributed to 50 ~ 86% of days exceeding the Chinese
national air quality standards (Grade II, ~75 ppb) in the whole October (Fig. 1c). Other regions exhibit less
MDA8 $O_3$ increases, which are 20%, 16% and 3% for East, North and West China, respectively (Fig. 1b).
Negligible MDA8 $O_3$ increase in West China is consistent with vast rural areas and less anthropogenic
impacts. This result suggests that changes in anthropogenic emissions have significant impacts on MDA8
$O_3$ during the CNDH in the South, East, and North China, similar to a previous observation study (Xu et
al., 2017).

Nine key cities are then selected for analyzing the causes and impacts of the remarkable MDA8 $O_3$

rises. Comprehensive criteria were adopted in selection according to: (1) acute MDA8 $O_3$ increases (e.g.,
Changsha and Shenzhen), and (2) important provincial capitals (e.g., Beijing and Shanghai) and famous
tourist cities (e.g., Sanya). The selected key cities are delegates of broad regions in China except for West
China (Fig. S1) with insignificant MDA8 $O_3$ increase (Table S3) and fewer traveling cities. The MDA8 $O_3$
increased by 48±37 % during the 2018 CNDH in these key cities. The highest MDA8 $O_3$ is observed in
Zhuhai, reaching 98 ppb on average with the peak of 107 ppb. The MDA8 $O_3$ in Sanya increases twofold
compared to PRE-CNDH. This is unexpected as Sanya is less-concerned regarding air pollution and known
for less anthropogenic emissions (Wang et al., 2015). Other key cities show 8-70% increases during the
CNDH. The exact causes of substantial $O_3$ increases in these cities are of high interest and explored below.



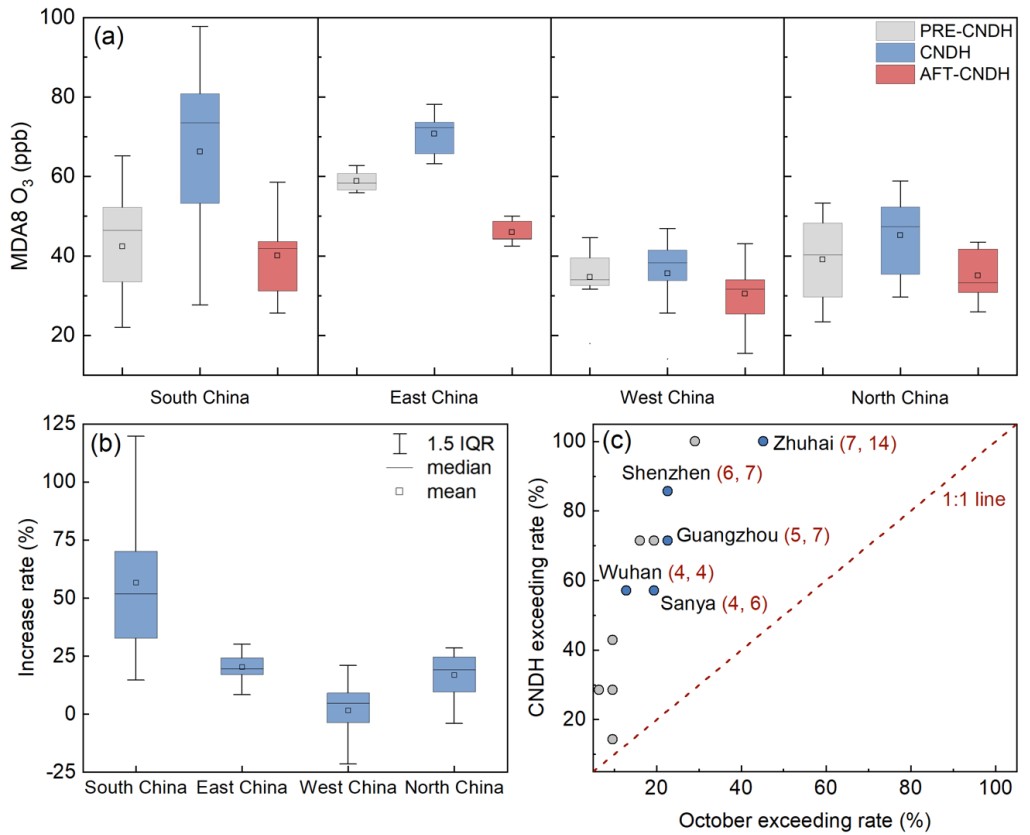


**Figure 1. (a)** The observed average MDA8 $O_3$ in PRE-CNDH, CNDH and AFT-CNDH in South, East,

West and North China in 2018; **(b)** The increase rate of observed MDA8 $O_3$ during CNDH; **(c)** The

exceeding rate of observed MDA8 $O_3$ in CNDH and October. Blue dots refer to the key cities and grey

dots represent other cities. The pairs of values in the parentheses following city name are the exceeding

days in CNDH and October, respectively. IQR is the interquartile range.

### 3.2 Increased $O_3$ precursor emissions during CNDH

The CMAQ is capable to represent the changes in observed MDA8 $O_3$ (Fig. 2). Generally, increasing

trends of MDA8 $O_3$ are found in vast areas from PRE-CNDH to CNDH, suggesting the elevated $O_3$ occurs

on a regional-scale. In South China, MDA8 $O_3$ reaches ~90 ppb that is approximately 1.2 times of the Class

II standard with averaged increase rate of 30%. In contrast, the highest MDA8 $O_3$ drops sharply to 60 ppb

in same regions in AFT-CNDH. High $O_3\_NO_x$ and $O_3\_VOC$ levels are also found during CNDH with

different spatial distributions (Fig. 2). The rising $O_3\_NO_x$ areas are mainly located in South China covering





Hubei, Hunan, Guangxi, Jiangxi, north Guangdong and Fujian provinces with average increase of ~5-10
ppb. While high $O_3\_VOC$ regions are in developed city clusters such as the NCP, YRD and PRD regions.
In the PRD, peak $O_3\_VOC$ is over 30 ppb during the CNDH, which is 1.5 times of that in PRE-CNDH.
Similar to MDA8 $O_3$, decreases in both $O_3\_NO_x$ and $O_3\_VOC$ are found in AFT-CNDH. For the nine key
cities, $O_3\_NO_x$ and $O_3\_VOC$ are also increased during CNDH. In Sanya, non-background $O_3$ during CNDH
is two times of that in PRE-AFDH. The peak of non-background $O_3$ ($O_3\_NO_x$ + $O_3\_VOC$) is over 80 ppb
in Beijing and Zhuhai, indicating that $O_3$ formation plays an important role during CNDH (Fig. 3). In
megacities such as Beijing, $O_3\_VOC$ is the major contributor to elevated $O_3$, while $O_3\_NO_x$ becomes
significant in tourist cities such as Sanya.

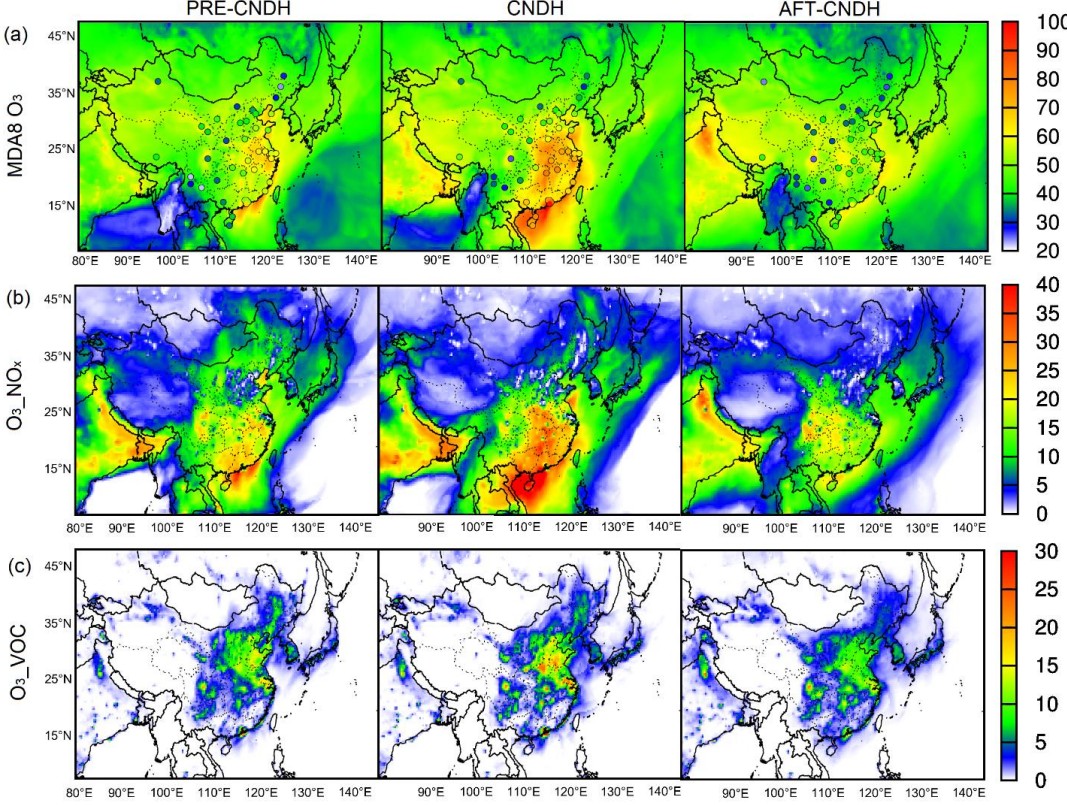

**Figure 2. (a)** Comparison of observed (circle) and predicted MDA8 $O_3$; **(b)** Spatial distribution of $O_3\_NO_x$;
**(c)** Spatial distribution of $O_3\_VOC$ in China in PRE-CNDH, CNDH and AFT-CNDH, respectively. Units
are ppb.

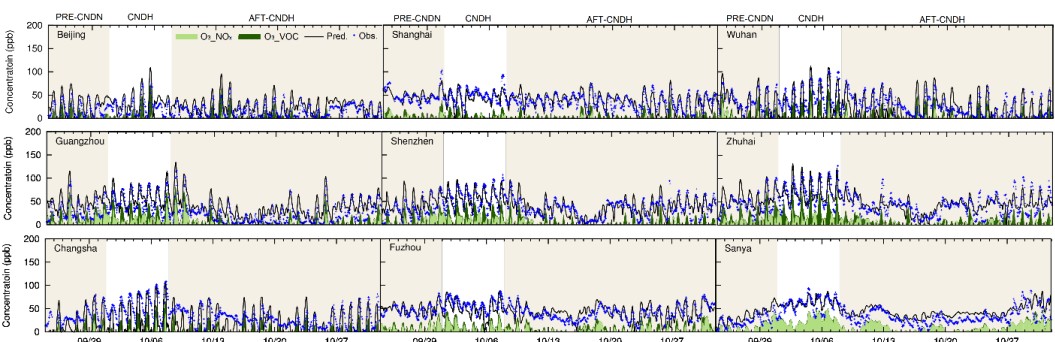

**Figure 3.** Hourly O$_3$ and its source apportionment results in nine key cities.

Considering O$_3$ sensitivity regimes (determined by equation (S1)), no noticeable differences are observed between PRE-CNDH and CNDH (Fig. S3). During CNDH, the VOC-limited regions are mainly in the NCP and YRD accompanied by high O$_3$_VOC. In South China, O$_3$ formation is under the transition regime in most regions and NO$_x$-limited regions are in Fujian as well as part of Guangdong and Guangxi. Increasing NO$_x$ emissions are observed in Guangxi and Guangdong with relative increase of up to 100% during CNDH, corresponding to rising O$_3$ in these NO$_x$-limited regions (Fig. S4). Simultaneously, higher anthropogenic VOC emissions are observed during CNDH in South China, leading to elevated O$_3$ in the transition regime when O$_3$ formation is controlled by both VOC and NO$_x$. In contrast, during AFT-CNDH, more areas turn into transition regime in South China. The decreases in biogenic VOCs (compared to CNDH) (Fig S4) due to temperature (Fig. S5) reduce MDA8 O$_3$ for regions in transition regime during AFT-CNDH. Accordingly, changes in O$_3$ highly depends on its precursor emissions (NO$_x$ and VOCs) as well as the sensitivity regime.

Transportation increase due to tourism is also a potential source of elevated O$_3$ during holidays (Xu et al., 2017). However, changes in transportation emissions are not considered in this study due to lack of related statistical data. Residents prefer to travel during CNDH and thus more important impacts may be from mobile sources (Zhao et al., 2019). Traveling by private cars is the most common approach, leading to a significant increase in vehicle activities (Wang et al., 2019c). Time-varying coefficients are estimated to describe traffic flow according to AMAP (2018) report during 2018 CNDH (Fig. S6). On average, CNDH is 2.2 times the traffic flow of ordinary weeks. The heavy traffic flow occurs on October 1st (coefficient of 16.3%) and 5th (6.1%), due to intensive departure and return, respectively. Hourly variations of traffic flow in CNDH are similar to weekends, having a flatter trend compared to workdays (Liu et al., 2018b). A real-time vehicle emission inventory should be developed in future to better predict O$_3$ changes during CNDH.





### 3.3 Impacts of regional Transport during CNDH

Regional transport is also a major contributor to enhanced MDA8 $O_3$ during CNDH. The higher $O_3$ production rates (increase rate up to ~150%) are observed mainly in the urban regions (the NCP, YRD, and PRD) in China (Fig. S7). With north winds (Fig. S5), $O_3$ is transported from the north regions to downwind South China to cause aggravated $O_3$. In the nine key cites, enhanced regional transport (HADV: horizontal advection) of $O_3$ in Beijing, Changsha, Fuzhou, Shenzhen, Sanya, and Shanghai is as high as 90 ppb (Fig. S8).

A regional-source tracking simulation was conducted in the PRD that occurred important $O_3$ elevation to qualify the impacts of regional transport. The emissions were classified into 7 regional types (Fig. S9): the local PRD (GD), northern part (NOR), southern part (SOU), central part (CEN), western part (WES), southeast part (SWE) and other countries (OTH). The detailed model description could be found in Wang et al. (2020a). Although the local sector contributes more than 50% non-background $O_3$ from PRE-CNDH to AFT-CNDH the more significant $O_3$ regional transport are predicted during the late PRE-CHDH and CNDH in the PRD, manifesting important role of the its important role in the $O_3$ elevation (Fig. 4 and S10). The SOU sector is the most crucial contributors among all these regional sectors outside Guangdong due to the prevailing south wind.

In these PRD key cities (Guangzhou, Shenzhen, and Zhuhai), the contribution of SOU sector in the non-background $O_3$ is up to ~30 ppb, mainly occurring in the nighttime and early morning (Figure 4). In the noontime, ~10-15% non-background $O_3$ is from SOU sector during the CNDH compared to less 5% in other periods. The $O_3\_NO_x$ shows more significant regional transport characteristics than that of the $O_3\_VOC$ (Fig. S11 and S12). During the late pre-CNDH and the CNDH, the contribution from regional transport in the $O_3\_NO_x$ is up to 35 ppb. Due to the enhanced regional transport during the CNDH, the $O_3\_NO_x$ could be even transported from the long-distance sector as NOR to the PRD. The peak of $O_3\_NO_x$ due to the regional transport is predicted in the midnight, which is different of that of $O_3\_VOC$ (peak in the noontime).

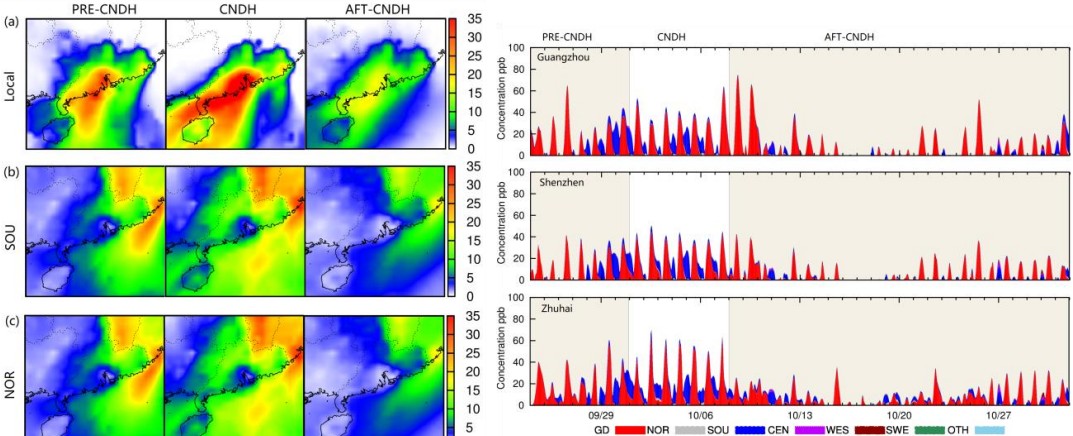

**Figure 4. (a)** Average regional contributions to non-background $O_3$ in from the PRD local emissions and

emissions in SOU, and NOR sectors and **(b)** regional contributions from all sectors to non-background $O_3$

in the PRD key cities (Guangzhou, Shenzhen, and Zhuhai) during the simulation periods.

**3.4 Aggravated Health Risk during CNDH**

It is recognized that $O_3$ pollution induces serious health risks from CVD, RD, COPD, hypertension

and stroke (Lelieveld et al., 2013;Yin et al., 2017;Huang et al., 2018;Krewski et al., 2009). Elevated MDA8

$O_3$ during CNDH leads to significantly higher health risks (Fig. 5). Estimated total national daily mortality

(from all non-accidental causes) due to MDA8 $O_3$ is 2629 during CNDH, 33% higher than that (1982) in

PRE-CNDH. All above $O_3$-related diseases have noticeable increases in national daily mortality during

CNDH. The highest health risk among these diseases is from CVD (674 during the CNDH), which is

consistent with Yin et al. (2017), followed by RD (219), COPD (213), hypertension (189) and stroke (22).

The COPD mortality due to $O_3$ in this study is comparable with 152-220 in Liu et al. (2018a). In AFT-

CNDH, total daily mortality (drops to 1653) and mortality from all diseases decreases due to substantial $O_3$

reduction. Also, a substantial increase in the total daily mortality is shown throughout China during the

CNDH especially at those densely-populated regions (e.g., the YRD and PRD) (Fig. S11), which is

consistent with previous studies (Chen et al., 2018;Liu et al., 2018a;Wang et al., 2020b).



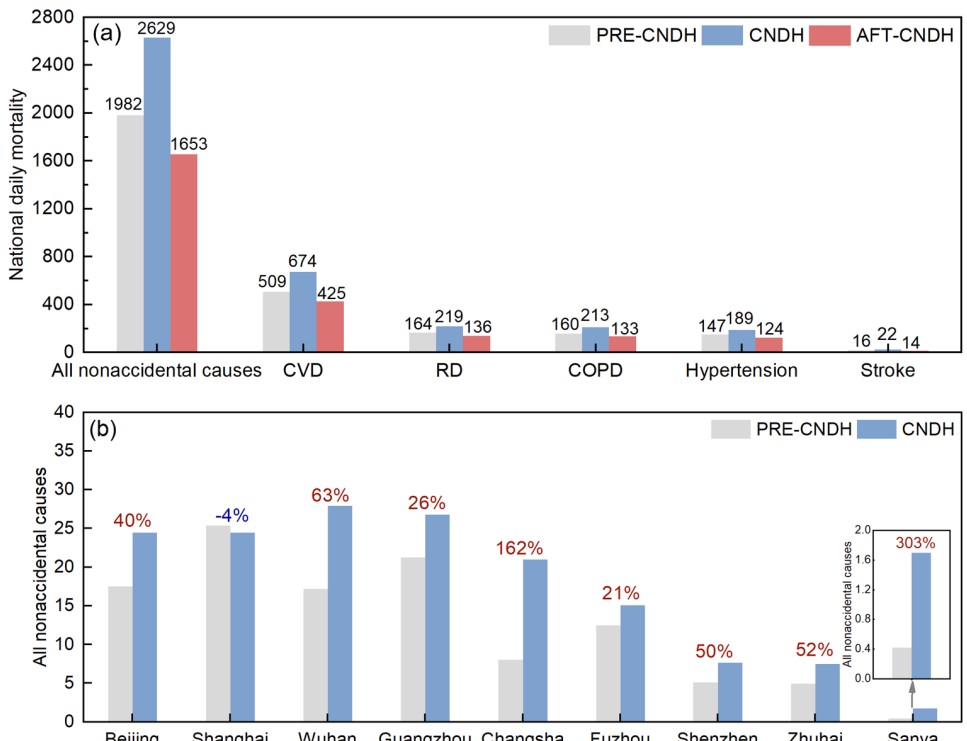

**Figure 5. (a)** National daily mortality from all non-accidental causes, CVD, RD, COPD, hypertension and stroke attributed to O$_3$ in RRE-CDNH, CNDH and AFT-CNDH and **(b)** Daily mortality from all non-accidental causes due to O$_3$ in the 9 key cities. Red/blue values above the bars are the increase/decrease rates of daily mortality from PRE-CNDH to CNDH. CVD: cardiovascular diseases; RD: respiratory diseases; COPD: chronic obstructive pulmonary disease.

Considering the nine key cities, total daily mortality increases from PRE-CNDH to CNDH except Shanghai, in which O$_3$ is slightly underestimated. Four megacities (Beijing, Shanghai, Wuhan and Guangzhou) with enormous populations have the highest daily deaths (24-28) during CNDH, 50% larger than the mean level (16) in the other 272 Chinese cities (Chen et al., 2018;Yin et al., 2017). It is worth noting that higher increase rate of daily mortality is found in tourist cities (Sanya and Changsha). In Sanya, daily deaths even increase by as high as 303% from PRE-CNDH to CNDH. An even higher increase in health risk may occur in Sanya if consider sharply increased tourists flow during CNDH.


**4. Conclusion and Implications**

In this study, we find the significant increase in $O_3$ during the CNDH throughout China especially

in the south part, which is attributed to the changes in precursor emissions, sensitivity regime, and the
enhanced regional transport. Moreover, the elevated $O_3$ also causes serious impacts on human health, with
total daily mortality from all non-accidental causes increasing from 151 to 201 in China. More
comprehensive studies should be conducted to better understand the long-holiday impacts (such as during
the CNDH) of $O_3$ in the future and here we suggest:
1)    More strident emission control policies should be implemented in China before and during

CNDH to inhibit the elevated $O_3$. And more localized control policies with the consideration of

the $O_3$ sensitivity regimes should be applied.

2)    For reducing the health risk from the elevated $O_3$, it is suggested to avoid travelling in rush hours

especially in midday during the CNDH.

3)    Reducing the activities of private gasoline vehicles is effective to mitigate excess emissions

during the CNDH. It is encouraged to go out by electric car or public transportation such as bus,

subway and train.


*Acknowledgments.* This work was supported by the National Natural Science Foundation of China (Grant
No. 41530641/41673116/41961144029), the National Key Research and Development Program
(2017YFC02122802), Theme-based Research Scheme (No. T24-504/17-N), Youth Innovation Promotion
Association, CAS (2017406) and Guangdong Foundation for Program of Science and Technology Research
(Grant No. 2017B030314057).
*Author contributions.* PW and YZ designed the research. PW, JS, MX, SS and HZ analyzed the data. PW
performed air quality model. PW and YZ wrote the manuscript with comments from all co-authors.
*Competing interests.* The authors declare that they have no conflict of interest.
*Data availability.* The datasets used in the study can be accessed from websites listed in the references or
by contacting the corresponding author (zhang_yl86@gig.ac.cn).









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
