# Peer review of "Unexpected enhancement of ozone exposure and health risks during the Chinese National Day Holiday"

_Atmospheric Chemistry and Physics, 2020_

## Author Comment (AC2)

**Comments Response**

**Journal: Atmospheric Chemistry and Physics**

**Manuscript ID: acp-2020-1302**

**Title: "Unexpected enhancement of ozone exposure and health risks during National Day in China"**

Dear Referee #2,

We appreciate your comments to help improve the manuscript and tried our best to address your comments. The detailed responses and related changes are shown in below. Our response is in blue and the modifications in the manuscript are in red. All figures are included in the attached PDF file.

This manuscript investigates the causes of high O3 episode during Chinese National Day Holiday using CMAQ modeling. The high O3 concentration is found to be caused by enhanced anthropogenic emissions and regional transport. Further, the health risks of these high O3 episode are estimated based on the response function of premature mortality of O3 exposure. The scope of this manuscript is of interest and fits the Atmospheric Chemistry and Physics journal. However, the illustration of the manuscript makes it a bit difficult to review fairly. The readability can be easily improved by elaborating several key terminologies and large fonts in figures. For example, two terms "O3_VOC" and "O3_NOx" are discussed throughout the manuscript to diagnose the O3 chemistry, but they are not clearly defined in the manuscript. The font of figure 3 is too small. The color scheme in figure 4b is impossible to read. I believe this study is publishable, but requires substantial revisions.

**Response:** Thanks for the comments. In the revision, we tried our best to modify our manuscript, including the related figures (Figure 3 and Figure 4b) to improve our study. Below is the response to each specific comment.

1. The importance of regional transport. Look at Figure 2a CNDH, the O3 concentration is up to 100 ppb in south China sea around Hainan and it is higher than the mainland China. Is this real? If so, what's the impact of such high O3 concentration on southern China? Is this the major cause of the high O3 episode during CNDH? Figure S5 suggests the prevailing wind direction is from mainland to ocean during CNDH in CMAQ. Is this consistent with local measurements? Line 217 indicates the south wind is prevailing. I am confused.

**Response:** From Figure 2a and Figure S5, the high $O_3$ in the sea around Hainan is mainly due to the regional transport under the impact of the north wind. The $O_3$ observation data in the south China sea is not available, so we couldn't further evaluate the $O_3$ level in the ocean. While Table S5 and Figure 2a showed our predicted $O_3$ agreed well with the observed $O_3$ throughout China, which could provide robust results for the air quality analysis. Our study also concluded that the regional transport corresponds to the elevated $O_3$ during the CNDH (in section 3.3). In addition, we also compared the observed and predicted wind field in the key cities (Guangzhou and Shenzhen; Zhuhai's data was not available) in the attached PDF file (Figure R2-1 and R2-2). It is shown in Figures R2-1 and R2-1 that the prevailing wind direction is north, which is consistent with the prediction. The line 217 should be "north wind". Sorry for the mistake and we have corrected it in the revised manuscript.

**Changes in manuscript:** (line 236-238) The SOU sector is the most crucial contributor among all these regional sectors outside Guangdong due to the prevailing north wind.

[Figure]

**Figure R2-1.** The observed wind field in Guangzhou during the CNDH.

[Figure]

**Figure R2-2.** The observed wind field in Shenzhen during the CNDH.

2. Line 24. This "303%" overstates the health risk, because the absolute difference is small (0.4 vs 1.6 in Figure 5).

**Response:** Thanks for the comment. We used the absolute number in the revised manuscript instead of the increasing, which may avoid the 'overestimation' of the health risk. The aggravated health risk in the tourist cities such as Sanya is a crucial message for our study, and we prefer to keep this message in the abstract.

**Changes in manuscript:** (line 23-24) Moreover, in tourist cities such as Sanya, daily mortality even increases significantly from 0.4 to 1.6.

3. The following terminologies/calculations should be elaborated: O3_VOC, O3_NOx, O3 production rate, and exceeding rate (Figure 1c).

**Response:** Thanks for the comment. We have elaborated these terminologies in the manuscript and corresponding figures caption.

**Changes in manuscript:** (line 76-80) Two non-reactive $O_3$ species: $O_3\_NO_x$ and $O_3\_VOC$ are added in the CMAQ model to quantify the $O_3$ attributable to $NO_x$ and VOCs, respectively. In particular, $O_3\_NO_x$ stands for the $O_3$ formation is under $NO_x$-limited control, and $O_3\_VOC$ stands for the $O_3$ formation is under VOC-limited control. The details of the 3R scheme and the calculation of $O_3\_NOx$ and $O_3\_VOC$ are described in *Wang et al.* [2019]. (line 157-160, caption of Figure 1) Figure 1. (a) The observed average MDA8 $O_3$ in PRE-CNDH, CNDH and AFT-CNDH in South, East, West and North China in 2018; (b) The increase rate of observed MDA8 $O_3$ during CNDH; (c) The exceeding rate of observed MDA8 $O_3$ in CNDH and October (the exceeding days during the CNDH divided by that during the October, exceeding_CNDH/exceeding_October). (line 181-183, caption of Figure 2) Figure 2. (a) Comparison of observed (circle) and predicted MDA8 $O_3$; (b) Spatial distribution of $O_3\_NO_x$; (c) Spatial distribution of $O_3\_VOC$ in China in PRE-CNDH, CNDH and AFT-CNDH, respectively. Units are ppb. $O_3\_NO_x$ and $O_3\_VOC$ are the $O_3$ attributed to $NO_x$ and VOCs, respectively. (caption of Figure S7 in the supplement) The $O_3$ production rates stand for the total production of $O_3$ by adding all reactions that $O_3$ is defined as a product. (line 222-224) The higher $O_3$ production rates that are calculated by the PA process directly in the CMAQ model (increase rate up to ~150%) are predicted observed mainly in the urban regions (the NCP, YRD, and PRD) in China (Fig. S7).

4. To corroborate the estimated health risks, the estimated daily mortality (non-accidental causes) should be compared to real mortality data, if possible.

**Response:** Thanks for the comment. The real mortality data is not available although we have tried our best to find this data.

**Reference:**

Wang, P., Y. Chen, J. Hu, H. Zhang, and Q. Ying (2019), Attribution of Tropospheric Ozone to NOx and VOC Emissions: Considering Ozone Formation in the Transition Regime, *Environmental Science & Technology*, *53*(3), 1404-1412, doi:10.1021/acs.est.8b05981.

---

## Author Response (AR1)

**Comments Response**

Journal: Atmospheric Chemistry and Physics

Manuscript ID: acp-2020-1302

Title: "Unexpected enhancement of ozone exposure and health risks during National Day in China"

**Dear Referee #1,**

We appreciate your comments to help improve the manuscript and tried our best to address your comments. The detailed responses and related changes are shown in below. Our response is in blue and the modifications in the manuscript are in red. All figures are included in the attached PDF file.

**General:**

The manuscript presents a topical research, i.e. to understand the elevated O3 issue in China due to the holiday impact. This study reported that the drastically rising MDA8 O3 were observed during the CNDH with the increasing rate up to 120% even in some pristine regions, which also induced 33% additional deaths through China. It was shown that increased precursor emissions and regional transport were corresponding to the O3 elevation. This is the first comprehensive study to investigate O3 pollution during CNDH at national level and could provide useful suggestion for the policy makers. The manuscript is easy to follow and fit to the scope of ACP very well. I have some minor comments below for the authors to address.

**Response:** Thanks for the recognition of our study. Below is the response to each specific comment. **Minor comments:**

**Line 90~91: Could the author explain more for the IPR and PA tools in the CMAQ model?**

Response: The IPR (integrated process rate analysis) and IRR (integrated reaction rate analysis) are all defined as the
PA (process analysis) in the CMAQ model(https://www.cmascenter.org/cmaq/science\_documentation/pdf/ch16.pdf+&cd=1&hl=zh-CN&ct=clnk). PA aims to
provide quantitative information on the process of the chemical reactions and other atmospheric processes that are
being simulated, illustrating how the CMAQ model calculated its predictions. The IPR analysis quantifies the relative
contributions of individual atmospheric physical and chemical processes in the CMAQ model.

**Changes in manuscript:** (lines 95-100) In the CMAQ model, the IPR and integrated reaction rate analysis (IRR) were all defined as the PA. PA aims to provide quantitative information on the process of the chemical reactions and other atmospheric processes that are being simulated, illustrating how the CMAQ model calculated its predictions. The IPR was used to determine the relative contributions of individual atmospheric physical and chemical processes in the CMAQ model.

Line 135~136: as readers may not be familiar with West China, please add a reference to show that West China has less anthropogenic impacts.

**Response:** Thanks for the comments. We've added a related reference to show the less anthropogenic impact of West China.

**Changes in manuscript:** (lines 141-142) Negligible MDA8 O3 increase in West China is consistent with vast rural areas and less anthropogenic impacts (Wang et al., 2017).

Line 147~148: it should be mentioned that MDA8 O3 in Shanghai during the CNDH slightly decreased compared with that before CNDH.

**Response:** We have checked the observation data and confirm that the MDA8  $O_3$  in Shanghai increased from 58.3 ppb to 63.2 ppb during the CNDH (Table S3). The reviewer may refer to the graphical abstract, which shows decreased total daily mortality in Shanghai but not decreased  $O_3$  levels. The model simulation slightly underestimates the observed  $O_3$  levels in Shanghai during the CNDH, which causes the decreased total daily mortality. We have re-arraged the related content to better clarify this point in Line 273-274.

**Changes in manuscript:** (lines 273-274) Except for Shanghai (in which  $O_3$  is slightly underestimated), the other eight key cities increased their total daily mortality rates from PRE-CNDH to CNDH.

Line 188: could the author explain more about meteorology impacts such as the variation of the temperature on the O3 during the CNDH?

Response: Thanks for the comment. We've added more analysis of the meteorology impacts in section 3.3.

**Changes in manuscript:** (line 218-224) Regional transport is also a significant contributor to enhanced MDA8  $O_3$  during CNDH. As shown in Fig. S5, the lower temperature is predicted during the CNDH compared to the PRE-CNDH. In PRD, the average temperature drops from 25 °C to 23 °C, leading to a lower  $O_3$  level in previous studies (Fu et al., 2015;Bloomer et al., 2009;Pusede et al., 2015). Meanwhile, the increasing wind speed is predicted in the PRD, which is able to facilitate regional transport. The higher  $O_3$  production rates that are calculated by the PA process directly in the CMAQ model (increase rate up to ~150%) are predicted mainly in the urban regions (the NCP, YRD, and PRD) in China (Fig. S7).

Line 195: Could the author discuss how will the coefficients from the AMAP be applied in the emission inventory?

**Response:** In the future study, we consider using the real-time coefficients from the AMAP to adjust the traffic emissions. First, an average emission adjustment factor from AMAP will be applied in the simulation during the CNDH to investigate the impacts on  $O_3$  throughout China. And then, a daily or even hourly adjustment factor (if possible) will be applied in the transport emission. In addition, the localized real-time traffic flow data will be considered (if available) as well as the coefficients from AMAP, aiming to reflect the emission variations during the CNDH on a regional scale. By including the localized real-time data, we will be capable of conducting a more compressive study of the emission changes of the traffic sector during the CNDH.

Line 228: please label the key cities in the PRD in the Figure 4

**Response:** We have labeled the key cities (GZ: Guangzhou, SZ: Shenzhen, and ZH: Zhuhai) in the revised Figure 5 (Also shown in the figure below R1-1).

**Figure R1-1.** (a) Average regional contributions to non-background  $O_3$  from the PRD local emissions and emissions in SOU, and NOR sectors and (b) regional contributions from all sectors to non-background  $O_3$  in the PRD key cities (Guangzhou, Shenzhen, and Zhuhai) during the simulation periods. GZ: Guangzhou, SZ: Shenzhen, and ZH: Zhuhai.

**References:**

Bloomer, B. J., Stehr, J. W., Piety, C. A., Salawitch, R. J., and Dickerson, R. R.: Observed relationships of ozone air pollution with temperature and emissions, Geophysical Research Letters, 36, 2009.

Fu, T.-M., Zheng, Y., Paulot, F., Mao, J., and Yantosca, R. M.: Positive but variable sensitivity of August surface ozone to large-scale warming in the southeast United States, Nature Climate Change, 5, 454-458, 2015. Pusede, S. E., Steiner, A. L., and Cohen, R. C.: Temperature and recent trends in the chemistry of continental surface ozone, Chemical reviews, 115, 3898-3918, 2015.

Wang, J., Zhao, B., Wang, S., Yang, F., Xing, J., Morawska, L., Ding, A., Kulmala, M., Kerminen, V.-M., Kujansuu, J., Wang, Z., Ding, D., Zhang, X., Wang, H., Tian, M., Petäjä, T., Jiang, J., and Hao, J.: Particulate matter pollution over China and the effects of control policies, Science of The Total Environment, 584-585, 426-447, https://doi.org/10.1016/j.scitotenv.2017.01.027, 2017.

**Comments Response**

Journal: Atmospheric Chemistry and Physics

**Manuscript ID: acp-2020-1302**

Title: "Unexpected enhancement of ozone exposure and health risks during National Day in China"

**Dear Referee #2,**

We appreciate your comments to help improve the manuscript and tried our best to address your comments. The detailed responses and related changes are shown in below. Our response is in blue and the modifications in the manuscript are in red. All figures are included in the attached PDF file.

This manuscript investigates the causes of high O3 episode during Chinese National Day Holiday using CMAQ modeling. The high O3 concentration is found to be caused by enhanced anthropogenic emissions and regional transport. Further, the health risks of these high O3 episode are estimated based on the response function of premature mortality of O3 exposure. The scope of this manuscript is of interest and fits the Atmospheric Chemistry and Physics journal. However, the illustration of the manuscript makes it a bit difficult to review fairly. The readability can be easily improved by elaborating several key terminologies and large fonts in figures. For example, two terms "O3\_VOC" and "O3\_NOx" are discussed throughout the manuscript to diagnose the O3 chemistry, but they are not clearly defined in the manuscript. The font of figure 3 is too small. The color scheme in figure 4b is impossible to read. I believe this study is publishable, but requires substantial revisions.

**Response:** Thanks for the comments. In the revision, we tried our best to modify our manuscript, including the related figures (Figure 3 and Figure 4b) to improve our study. Below is the response to each specific comment.

1. The importance of regional transport. Look at Figure 2a CNDH, the O3 concentration is up to 100 ppb in south China sea around Hainan and it is higher than the mainland China. Is this real? If so, what's the impact of such high O3 concentration on southern China? Is this the major cause of the high O3 episode during CNDH? Figure S5 suggests the prevailing wind direction is from mainland to ocean during CNDH in CMAQ. Is this consistent with local measurements? Line 217 indicates the south wind is prevailing. I am confused.

**Response:** From Figure 2a and Figure S5, the high  $O_3$  in the sea around Hainan is mainly due to the regional transport under the impact of the north wind. The  $O_3$  observation data in the south China sea is not available, so we couldn't further evaluate the  $O_3$  level in the ocean. While Table S5 and Figure 2a showed our predicted  $O_3$  agreed well with the observed  $O_3$  throughout China, which could provide robust results for the air quality analysis. Our study also concluded that the regional transport corresponds to the elevated  $O_3$  during the CNDH (in section 3.3). In addition, we also compared the observed and predicted wind field in the key cities (Guangzhou and Shenzhen; Zhuhai's data was not available) in the attached PDF file (Figure R2-1 and R2-2). It is shown in Figures R2-1 and R2-1 that the prevailing wind direction is north, which is consistent with the prediction. The line 217 should be "north wind". Sorry for the mistake and we have corrected it in the revised manuscript.

**Changes in manuscript:** (line 236-238) The SOU sector is the most crucial contributor among all these regional sectors outside Guangdong due to the prevailing north wind.